# Light Intensity Regulates Low-Temperature Adaptability of Tea Plant through ROS Stress and Developmental Programs

**DOI:** 10.3390/ijms24129852

**Published:** 2023-06-07

**Authors:** Xin Zhang, Keyi Liu, Qianhui Tang, Liang Zeng, Zhijun Wu

**Affiliations:** 1College of Food Science, Southwest University, Chongqing 400715, China; 2Chongqing Key Laboratory of Speciality Food Co-Built by Sichuan and Chongqing, Southwest University, Chongqing 400715, China; 3Integrative Science Center of Germplasm Creation, Southwest University, Chongqing 401329, China; 4Tea Research Institute, Southwest University, Chongqing 400715, China

**Keywords:** tea plant, low temperature, light intensity, ROS, tissue development

## Abstract

Low-temperature stress limits global tea planting areas and production efficiency. Light is another essential ecological factor that acts in conjunction with temperature in the plant life cycle. However, it is unclear whether the differential light environment affects the low temperature adaptability of tea plant (*Camellia* sect. *Thea*). In this study, tea plant materials in three groups of light intensity treatments showed differentiated characteristics for low-temperature adaptability. Strong light (ST, 240 μmol·m^−2^·s^−1^) caused the degradation of chlorophyll and a decrease in peroxidase (POD), superoxide dismutase (SOD), catalase (CAT), ascorbate peroxidase (APX), and polyphenol oxidase (PPO) activities, as well as an increase in soluble sugar, soluble protein, malondialdehyde (MDA), and relative conductivity in tea leaves. In contrast, antioxidant enzyme activities, chlorophyll content, and relative conductivity were highest in weak light (WT, 15 μmol·m^−2^·s^−1^). Damage was observed in both ST and WT materials relative to moderate light intensity (MT, 160 μmol·m^−2^·s^−1^) in a frost resistance test. Chlorophyll degradation in strong light was a behavior that prevented photodamage, and the maximum photosynthetic quantum yield of PS II (*Fv*/*Fm*) decreased with increasing light intensity. This suggests that the browning that occurs on the leaf surface of ST materials through frost may have been stressed by the previous increase in reactive oxygen species (ROS). Frost intolerance of WT materials is mainly related to delayed tissue development and tenderness holding. Interestingly, transcriptome sequencing revealed that stronger light favors starch biosynthesis, while cellulose biosynthesis is enhanced in weaker light. It showed that light intensity mediated the form of carbon fixation in tea plant, and this was associated with low-temperature adaptability.

## 1. Introduction

Light is a necessary ecological environmental factor for tea plant (*Camellia* sect. *Thea*) to survive. The effect of light on the economic traits of tea plants is multifaceted, such as leaf size, shoot weight, length of harvestable time, and formation of health compounds [1,2,3]. Some of the traits have been shown to be the behavior of tea plants actively adapting to the light environment. Tea plants improve light energy utilization by increasing their leaf area and chlorophyll content under low light [4,5]. In contrast, more flavonoids and lutein are formed in tea plants under a strong light to resist stress [6,7]. Suitable light conditions are considered to be an important prerequisite for a high yield and good quality of tea in production [5,8]. Limited by geographical location, climate, and vegetation, natural light conditions are not guaranteed to be optimal for tea production [9,10]. Therefore, the light environment of tea plants is usually regulated by some artificial means in cultivation, such as the use of shade nets and vegetation shading. Shading is effective in reducing strong light radiation, increasing photosynthetic efficiency, and avoiding photoinhibition in summer and autumn [9,11]. Pre-harvest shading is necessary for matcha, which will facilitate the formation of chlorophyll and theanine in tea leaves and reduce bitterness and astringency substances [12]. In addition, UV-B filtration and nighttime supplemental lighting devices in the exploratory stage may also be an option for tea production [13,14].

Low-temperature stresses, mainly including cold and frost damage, are one of the main meteorological factors limiting the expansion of tea plantation areas [15,16]. Cold damage inhibits the photosynthetic efficiency and shoot development of tea plants and triggers a decrease in the content of taste-related compounds such as water extracts, amino acids, and tea polyphenols in tea leaves [17,18]. Tea plants suffer from frost damage manifested as cell injury, browning, and even bud scorch [15,19]. These tea plants grown at high latitudes or high altitudes are faced with a higher risk of frost damage [20,21]. A number of satellite remote sensing techniques have recently been developed for assessing the risk of frost damage and economic loss to tea plants [20,21,22,23]. However, how to improve the low-temperature adaptation of tea plants and reduce the damage risk is still a difficult task. Factors that have been widely recognized as affecting the low-temperature adaptability of tea plants include cultivar, maturity, leaf structure, the duration of cold acclimation, and the metabolism of cold-resistant substances [24,25,26,27]. Generally, small- and mid-sized leaf cultivars are more resistant to cold than large-leaf cultivars of tea plants [25]. Leaf tissues with higher maturity or enriched cuticular wax were found to be more cold-tolerant [24,27]. Cold acclimation enhances cold resistance in tea plants and was shown to be related to the regulation of carbohydrate and antioxidant substance metabolism [26]. In addition, the molecular network ICE-CBF-COR and signaling processes in tea plants were revealed to play key roles in the cold resistance response [28].

The cultivation of tea plants has covered dozens of countries around the world and has become an important cash crop for farmers to increase their income [29,30]. In order to cope with the risk of climate anomalies and to meet the development requirements of the tea industry, the precise management, scientific planting, and sustainable development of tea plants are a few areas that need to be improved urgently [31,32,33]. However, the premise of these improvements in cultivation is the basis for the studies of phenomena and mechanisms involved in the environmental adaptation of tea plants. Due to the differences in latitude, altitude, slope, and orientation, the comprehensive conditions formed by the combination of ecological factors that tea plants are exposed to are actually complex [34,35]. Typically, the temperature decreases as the altitude rises, but the light intensity increases. Furthermore, the light intensity on the sunny side of the hillside is stronger than that on the shady side. Therefore, when tea plants are subjected to low-temperature stress, it may be accompanied by the effects of light conditions. Studies have shown that the adaptation of plants to low temperatures is mediated by light environments [36,37,38]. Moderate light intensity is thought to favor plant adaptation to low temperatures, while strong or weak light will exacerbate the damage caused by low-temperature stress [36,37,38]. The interference of these harsh light conditions with low-temperature tolerance in plants involves biological processes including damage from reactive oxygen species (ROS), photoinhibition, cell structure, and the metabolism of resistant substances [39,40]. In tea plant studies, low temperature and light environments are always treated as two separate objects, such that the experimental basis for whether low-temperature adaptability is related to light environments is unclear [16,25,41,42].

In this study, the effect of light intensity on the low-temperature adaptability of tea plants was assessed by artificially simulating light–temperature environments, as well as by combining phenotypic, chlorophyll fluorescence, cold resistance indicators, transcriptome sequencing, and frost resistance analyses. This study provides a new strategy for the research and application of improving the low-temperature adaptability of tea plants.

## 2. Results

### 2.1. Light Intensity Triggers Differences in Morphological Characteristics and Chlorophyll Fluorescence Parameters of Tea Plants

The tea plant leaf color varied significantly among the three groups of light intensity treatments (Figure 1a). Compared with MT samples, WT tea seedling leaves were dark green, while the leaf color of ST tea seedlings was more yellow. The tea plant leaf color showed a tendency to change from dark green to yellowish green with the increase in light intensity. Chlorophyll testing results showed that chlorophyll *a*, chlorophyll *b*, and the total chlorophyll of tea leaves decreased continuously with increasing light intensity (Figure 1b). It indicates that the chlorophyll content in tea leaves corresponded to the change in leaf color. Moreover, WT tea seedlings were weaker with thin and soft leaves compared to the MT samples, while ST tea seedlings were comparable in size and had stiffer leaves.

Chlorophyll fluorescence parameters of the three groups of materials were measured to analyze the effect of light intensity on light energy absorption and electron transfer in tea plant (Table 1). *Fm* was not significantly different between MT and WT, while it was significantly lower in ST. This indicates that the PS II reaction center chlorophyll *a* was affected by ST. *Fv*/*Fm* and *qP* represent the maximum photosynthetic quantum yield and the photochemical quenching of PS II, respectively. *Fv*/*Fm* decreased with increasing light intensity, indicating that the tea seedlings were susceptible to strong light stress, thus causing a decrease in photosynthetic activity. *qP* increased with increasing light intensity, which implies that ST promoted energy into photochemical processes. Actual photosynthetic efficiency (*Y(II)*) and electron transfer rate (*ETR*), and non-photochemical quenching (*NPQ*) were the highest in MT relative to WT and ST. These data reflect that moderate light intensity contributed to the enhancement of the photoprotective capacity and electron transfer effectiveness of tea seedlings.

### 2.2. Effect of Light Intensity on Low-Temperature Tolerance Indicators of Tea Plants

Some important physiological and biochemical indicators of samples, including MDA, soluble sugar, soluble protein, and antioxidant enzyme activities, were assayed to evaluate the effect of light intensity on the low-temperature tolerance of tea plant (Figure 2). The content of MDA, a product of membrane lipid peroxidation, increased with increasing light intensity (Figure 2a). Soluble sugar and soluble protein help to improve the cold tolerance of tea plant. The increase in soluble protein with increasing light intensity was slow, while the increase in soluble sugar was obvious (Figure 2b,c). In addition, five antioxidant enzymes, including APX (Figure 2d), CAT (Figure 2e), POD (Figure 2f), PPO (Figure 2g), and SOD (Figure 2h), decreased in activity with increasing light intensity. It showed that the scavenging abilities of antioxidant enzymes for ROS were weak in ST, while reducing the light intensity enhanced their activities.

### 2.3. Effect of Light Intensity on Frost Resistance of Tea Plants

The three groups of treatment materials showed morphological differences after frost treatment (Figure 3a,b). Tea plants in WT were most severely injured by frost, with curled leaf edges, slight scorching, and withered green leaves. Tea plants in ST were slightly frostbitten, with slightly curled leaf edges and brown spots on the leaf surface. Compared to WT and ST, the frost injury of tea leaves was not obvious in MT. Relative conductivity is an important indicator of plant cell membrane permeability. The permeability of the cell membrane will increase when it experiences damage from stress. The relative conductivities of the three groups of materials were less than 50% and increased with increasing light intensity before the frost resistance test (Figure 3c,d). This indicates that ST was prone to cause damage to the cell membrane of tea leaves in a non-frost environment, while reducing the light intensity could effectively protect the cell membrane. The relative conductivities of materials were again measured after frost treatment (−10 °C, 3 h). The relative conductivity in MT (<60%) is obviously lower relative to WT and ST (Figure 3b). The relative conductivity in WT was slightly higher than that in ST, and both were very close (>80%). This indicates that moderate light intensity is beneficial to the frost resistance of tea plant.

### 2.4. Transcriptome Sequencing and Differential Expression Gene (DEG) Analysis

The transcriptomes of samples were further sequenced and analyzed in order to understand the molecular response of tea plants in the three groups of treatments. The correlation of expression data between samples from transcriptome sequencing showed that the intra-group correlations were higher than inter-group correlations for almost all sequencing samples, indicating that the transcription levels in the three groups differed from each other (Figure 4a). DEGs obtained by threshold screening were presented in a Venn diagram (Figure 4b). In total, 3683 DEGs were obtained from WT vs. MT, while 1058 DEGs were found in ST vs. MT. ‘WT vs. MT DEG down’ and ‘ST vs. MT DEG up’ had the highest number of overlapping genes, while ‘WT vs. MT DEG up’ and ‘ST vs. MT DEG down’ had the lowest number of overlapping genes. This indicates that more up-regulated DEGs than down-regulated DEGs were induced with increasing light intensity.

KEGG annotation of transcriptome revealed that DEGs are involved in 126 metabolic pathways (Appendix A). The top 20 pathways are presented in Figure 4c,d based on the *p*-value ranking of the correlation between DEGs and metabolic pathways. The top pathways in WT vs. MT annotation were flavonoid biosynthesis, photosynthesis, tyrosine metabolism, glycolysis/gluconeogenesis, and circadian rhythm—plant. The number of annotated genes for glycolysis/gluconeogenesis, starch and sucrose metabolism, glutathione metabolism, pyruvate metabolism, and phenylpropanoid biosynthesis was higher (Figure 4c). The top pathways in WT vs. MT annotation were flavonoid biosynthesis, flavone and flavonol biosynthesis, and circadian rhythm—plant. The number of annotated genes for phenylpropanoid biosynthesis and glutathione metabolism was higher. In addition, we noted several genes involved in porphyrin metabolism (Figure 4d). Eleven of the top twenty pathways were shared between the two groups of annotations, showing that there were common features in the response of tea plants to light intensities. The obvious differences between the two groups of data are that sugar metabolism was regulated and involved more DEGs than flavonoid biosynthesis in WT vs. MT, whereas there are no pathways involving more DEGs than flavonoid biosynthesis in ST vs. MT.

### 2.5. Effect of Light Intensity on the Metabolism of Starch and Cellulose of Tea Plants

According to KEGG annotation results, sugar metabolism was regulated by light intensity, especially in WT vs. MT. To monitor the dynamics of sugar metabolism in tea plants as influenced by light intensity factors, the two main routes of sugar fixation, starch and cellulose biosynthesis, were investigated (Figure 5). Both starch and cellulose biosynthesis require sequential four-step catalytic reactions using Fru6P produced by the Calvin cycle as a substrate. Two branched routes are formed in the biosynthesis of starch and cellulose through the AGPase and UGPase catalysis of Glc1P, respectively (Figure 5a). A total of 65 expressed genes encoding eight enzymes involved in the starch and cellulose biosynthetic pathways were detected from the transcriptome (Figure 5b, Appendix A). The expression of some genes encoding PGM, AGPase, SS, and DBE was significantly up-regulated in ST. The expression of some genes encoding these enzymes was down-regulated in ST, but no significant differences were observed. Notably, most of the UGPase and CESA-encoding genes were up-regulated for expression in WT. The expression of about half of CsSSs, which are key genes for starch biosynthesis, did not appear to be associated with light intensity changes. These CsSSs, significantly down-regulated in ST, encode granule-bound starch synthases. The expression of 10 differentially expressed genes (DEGs) in three groups of light intensities was further detected by qRT-PCR (Figure 5c). The results showed that the relative expression levels of *CsPGM1*, *CsAGPase1*, *CsAGPase3*, *CsSS16*, *CsSS17*, *CsSS18*, and *CsSS19* tended to increase with increasing light intensity, and the relative expression level of *CsPGI3* was down-regulated in MT and ST. The relative expression levels of *CsAGPase2* and *CsDBE1* genes showed a trend of up-regulation followed by down-regulation. The qRT-PCR expression results of 10 DEGs essentially corresponded to the transcriptome data.

### 2.6. Effect of Light Intensity on Chlorophyll Degradation of Tea Plants

The chlorophyll content of plants is correlated with abiotic stress tolerance [43,44,45]. The test results showed that the stronger light caused the reduction in chlorophyll content and yellowing traits in tea leaves (Figure 1). Here, the effect of light intensity on chlorophyll degradation in tea leaves was further investigated (Figure 6). The catalysis of chlorophyll degradation in land plants starts with chlorophyll *b* reductase (CBR), including the isozymes NON-YELLOW COLORING 1 (NYC1) and NYC1-Like (NOL) (Figure 6a). A total of 13 expressed genes encoding six enzymes involved in chlorophyll degradation were detected from the transcriptome (Figure 6b, Appendix A). The expression of almost all of these coding genes was up-regulated with increasing light intensity. *CsSGR3* may be a special case with relatively high expression in WT. The expression of three DEGs under three groups of light intensities was further detected by qRT-PCR (Figure 6c). The results showed that the relative expression levels of *CsSGR1* and *CsRCCR* genes were significantly up-regulated with increasing light intensity, while the relative expression level of the *CsSGR4* gene was only significantly up-regulated in ST. The qRT-PCR expression results of 3 DEGs essentially corresponded to the transcriptome data.

## 3. Discussion

### 3.1. Moderate Light Intensity Improves Photosynthetic Activity of Tea Plants

Tea plant is shade-tolerant and even grows healthily in over-90%-sheltered environments [46,47]. In this study, the growth status of the tea plants remained good despite the use of 15 μmol·m^−2^·s^−1^ (WT). *Fv*/*Fm* value is usually used to assess the cold and frost resistance of plants, and a high *Fv*/*Fm* value in tea plants is considered to be an indication of excellent resistance [25,48]. *Fv*/*Fm* values of WT materials were the highest, predicting that the low light treatment was the least stressful for tea plants. Many plants, such as *Arabidopsis*, pepper, and lettuce, also experienced a decrease in *Fv*/*Fm* values due to stronger light [38,49,50]. This habit of adapting to low light allows tea plants to survive in the medium- and low-canopy environments of wild forests. Of course, the photosynthetic activity of WT materials was not optimal and lower than that of MT materials according to *qP*, *Y(II)*, *ETR*, and *NPQ* values. The appropriate increase in light intensity is beneficial to improve the photosynthetic activity of tea plants. In contrast, strong light is a different case for tea plants. Photoinhibition in tea plants caused by strong light was observed both in the laboratory and field [9,11]. Excess light energy is consumed by fluorescence and *NPQ* in higher plants to reduce oxidative damage [51,52]. ST materials have the highest *qP* values, indicating that more light energy enters the photochemical process. However, the *Y(II)*, *ETR*, and *NPQ* values were reduced in ST materials relative to those in MT materials, probably due to stress damage. Therefore, moderate light is generally more favorable to the photosynthetic activity of tea plants.

### 3.2. Moderate Light Intensity Enhances Frost Resistance of Tea Plants

The photosynthetic system of chloroplasts rapidly produces and accumulates ROS after plants are exposed to strong light [40]. Antioxidant enzymes in chloroplasts form the first defense to scavenge ROS (Figure 7). Several antioxidant enzyme activities were the highest in WT materials in this study, while these enzyme activities were the lowest in ST materials. Lower light was also found to increase antioxidant enzyme activities in tomato, wheat, and *Potamogeton crispus* [53,54,55]. Increased antioxidant enzyme activities are thought to facilitate the low-temperature adaptability of plants [56]. However, WT materials were not actually resistant to frost by the analysis of relative conductance and damage degree. This indicates that the frost resistance of WT materials may be dominated by factors other than high antioxidant enzyme activities. ST materials with the lowest antioxidant enzyme activities also exhibited a non-frost-resistant phenotype. The difference was that the leaf surface of ST materials was browned after frost damage. Previous studies have shown that the combination of low temperature and strong light exacerbates plant damage [38,39]. The relative conductivity and MDA of ST materials before frost were highest, which predicts that ST materials may have been damaged by strong light. Antioxidant enzyme activities have been applied to the assessment of low-temperature tolerance in tea plants [57,58,59,60]. This study revealed that light intensity affected the antioxidant enzyme activities and low-temperature adaptability of tea plants. Therefore, if the ambient light intensities are not uniform, the antioxidant enzyme activities are not reasonable as a basis for assessing the low-temperature tolerance of tea plants.

### 3.3. Light Intensity Affects the Balance of Starch and Cellulose Metabolism in Tea Plant

Starch acts as a carbon source pool in tea plant, and its metabolic processes are associated with low-temperature adaptability [26]. In cold acclimation, the accumulation of starch breakdown into soluble sugars helps to improve the adaptability of tea plant to low temperatures [26]. This study found that strong light could produce more soluble sugars and soluble proteins. The expression levels of many genes encoding starch synthesis-related enzymes, including SS, BE, and DBE, were up-regulated with increased light intensity. This predicts that the enhanced light intensity contributed to the starch accumulation and soluble sugar formation of carbon metabolism in tea plants. The weaker development of WT materials may be related to the relative lack of photosynthetic carbon fixation and carbon metabolism. According to previous field experience and study results, young shoots of tea plants are more susceptible to cold and frost damage [27]. This should be the main reason why WT materials are not frost resistant. Analysis of carbon metabolic branching revealed that the expression levels of many genes encoding key enzymes (UGPase and CESA) of cellulose biosynthesis trended to be up-regulated in WT materials. This indicates that monosaccharides produced from the Calvin cycle in WT materials may prefer to form cellulose for cytoskeleton building, rather than synthetic starch for storage (Figure 7). MT and ST materials may have preceded WT materials in the establishment of the cytoskeleton, transferring excess photosynthetic products through starch storage or soluble sugar formation.

### 3.4. Strong Light Causes Chlorophyll Degradation in Tea Leaves to Adapt to Light Stress

Numerous studies have shown that the chlorophyll content of tea leaves is decreased with increasing light intensity [4,6,7,61,62]. Furthermore, strong light promotes the degradation of chloroplasts in tea leaves [61]. This study found that the color of tea leaves differed significantly in the three groups of light treatments, and, similarly, the chlorophyll content was lower in stronger light intensities. Chlorophyll degradation under strong light has been revealed to be a light-adapted behavior [63]. In order to prevent photodamage, plants attenuate light-harvesting capacity by reducing the content of light-harvesting complex II (LHCII) [63]. Chlorophyll *b* degradation under strong light is catalyzed by CBR and triggers the degradation of LHCII [64]. The expression trends of CBR-encoded genes, *CsNYCs* and *CsNOLs*, in tea leaves were up-regulated in ST materials. Moreover, the expression levels of other genes in chlorophyll degradation metabolism, including *CsHCAR*, *CsSGRs*, *CsPPH*, *CsPAOs*, and *CsRCCR*, were almost up-regulated in ST materials as well. This indicates that these genes actively responded to light intensity and were involved in the degradation of chlorophyll under strong light (Figure 7). In cold resistance studies, the exogenous addition of melatonin and chitosan oligosaccharides alleviated the effects of cold stress by promoting antioxidant defense in tea plants [17,22,65]. Interestingly, the exogenous addition of these substances increased the chlorophyll content in tea leaves [17,22,65]. Chlorophyll content in leaves has been used as an important indicator of abiotic stress resistance in many plants [43,44,45]. Differences in cold resistance mechanisms between the light-sensitive and yellowing cultivar ‘Huangjinya’ and the normal leaf color cultivar ‘Yingshuang’ were found in tea plants [66,67,68]. Relative conductivity and MDA increased with increasing light intensity in this study and were negatively correlated with chlorophyll content. Therefore, the chlorophyll metabolism of tea plant is not only regulated by light intensity but also related to cold resistance.

## 4. Materials and Methods

### 4.1. Plant Material and Cold Treatment

The Nanchuan big tea tree (*C. nanchuanica*), an arboreal, large-leaved germplasm that is sensitive to light intensity and low temperatures, is an ideal material for testing [69]. The test materials for this study were 1-year-old seedlings of the Nanchuan big tea tree. Tea seedlings were planted in plastic pots with mixed peat moss and vermiculite (3:1, *v*/*v*) and placed on a light incubator for 7 d for pretreatment. Environmental conditions included 23 ± 1 °C temperature, 75% relative humidity, 160 μmol·m^−2^·s^−1^ light intensity, and a photoperiod of 16 h light per day. The culture temperature during darkness (8 h) was set to 5 °C to simulate a low-temperature environment. Light sources were full-spectrum LED lamps with visible wavelength bands (SananBio, Fujian, China).

Healthy tea seedlings were selected after pretreatment and divided into three groups and exposed to light intensities of 15 μmol·m^−2^·s^−1^ (weak light treatment, WT), 160 μmol·m^−2^·s^−1^ (moderate light treatment, MT), and 240 μmol·m^−2^·s^−1^ (strong light treatment, ST), respectively. Other experimental conditions were the same as the pretreatment conditions. Chlorophyll content, chlorophyll fluorescence, and conductivity were measured after 21 d. The shoots containing the first and second leaves were picked and immediately immersed in liquid nitrogen for subsequent analysis. In addition, the frost treatment of tea seedlings was performed in a low-temperature test chamber at −10 °C for 3 h. All samples were observed and tested by performing three biological and three technical replicates.

### 4.2. Determination of Chlorophyll Fluorescence

The chlorophyll fluorescence kinetic parameters of tea plants were determined by PAM-2500 (Walz, Effeltrich, Germany). The actinic light of PAM-2500 was red, the photosynthetically active radiation (PAR) measured by clip was 141 μmol·m^−2^·s^−1^, and the second leaf blades of tea seedlings were measured. The test leaves were dark-adapted by clips for 20 min. The measured chlorophyll fluorescence parameters included minimum fluorescence (*Fo*), maximum fluorescence (*Fm*), actual photosynthetic efficiency of PS II (*Y(II)*), photosynthetic electron transfer rate (*ETR*), non-photochemical quenching (*NPQ*), and photochemical quenching (*qP*). The variable fluorescence *Fv* (*Fv* = *Fm − Fo*) and the maximum photosynthetic quantum yield of PS II (*Fv*/*Fm*) were calculated.

### 4.3. Determination of Relative Conductivity

The relative conductivity of samples was determined by the soaking method. The second leaves of tea seedlings were picked, rinsed with deionized water to remove dirt, and then blotted with filter paper to dry the residual water on the leaf surface. Prepared leaves were punched with a hole punch avoiding the main leaf veins, and 6–8 mm diameter circular pieces were obtained. Five circular pieces were randomly selected from each sample and placed in a centrifuge tube with 10 mL of deionized water and placed at room temperature for 24 h. Extract conductivity was detected by DDS-307A (INESA, Shanghai, China). The first conductivity value was recorded as R_1_. The extract was heated by boiling water bath for 10 min, cooled to room temperature, and then shaken well to determine the conductivity, which was recorded as R_2_. Relative conductivity = R_1_/R_2_ × 100%.

### 4.4. Determination of Chlorophyll and Malondialdehyde (MDA) Content

The chlorophyll and malondialdehyde (MDA) contents of samples were extracted using Solarbio kits (Solarbio, Beijing, China). About 0.1 g of leaf tissue was weighed in a mortar containing 1 mL of distilled water and about 10 mg of reagent I, fully ground under low-light conditions, and then transferred to a 10 mL centrifuge tube. The extraction solution was prepared with anhydrous ethanol and acetone (1:2, *v*/*v*) and then used to rinse the mortar and fix the volume to 10 mL. The samples were wrapped with tin foil and extracted for 3 h. When the tissues were close to white, the samples were uploaded to ReadMax 1200 (Flash, Shanghai, China) for the determination of chlorophyll content. In addition, about 0.1 g of leaf tissue was weighed and placed in a mortar containing 1 mL of kit extraction solution and homogenized in an ice bath, followed by centrifugation at 8000× *g* for 10 min at 4 °C. The supernatant was transferred to a 2 mL centrifuge tube, and 3 times the volume of MDA analyzing solution was added and shaken well. The mixture was placed in a water bath at 100 °C for 60 min, cooled in an ice bath, and then centrifuged at 10,000× *g* for 10 min. The supernatant was uploaded to ReadMax 1200 for the determination of MDA. Three biological and three technical replicates were used for each treatment of samples.

### 4.5. Determination of Soluble Sugar, Soluble Protein, and Enzyme Activity

The assay samples for soluble sugar, soluble protein, and the activities of peroxidase (POD), superoxide dismutase (SOD), catalase (CAT), ascorbate peroxidase (APX), and polyphenol oxidase (PPO) were sent to Nanjing Convinced-test Technology Co., Ltd. (Nanjing, China). Soluble sugar and soluble protein were determined by the anthrone colorimetric method and colorimetric method using coomassie brilliant blue, respectively. The enzyme activities were determined by Molfarming kits (Molfarming, Nanjing, China). The enzyme units of POD and PPO were defined as A470 changing by 0.01 per min and A410 changing by 0.005 per min, respectively. The enzyme units of SOD, CAT, and APX were defined as the amount of nitroblue tetrazolium reduction rate inhibited by 50% per min, the amount of H_2_O_2_ decomposed by 1 μmol per min, and the amount of ascorbate oxidized by 1 μmol per min, respectively. Three biological and three technical replicates were used for each treatment of samples.

### 4.6. Transcriptomic Analysis

The total RNA of samples was extracted using a Quick RNA Isolation Kit (Huayueyang, Beijing, China). The RNA quality was evaluated via agarose gel electrophoresis, a NanoPhotometer (Implen, Munich, Germany), and an Agilent 2100 (Agilent, Santa Clara, CA, USA). The sequencing library was constructed by a NEBNext^®^ Ultra^TM^ RNA Library Prep Kit for Illumina^®^ (NEB, Ipswich, MA, USA), and sequence data (reads) were obtained by Illumina sequencing. HISAT2 (Johns Hopkins University, Baltimore, MD, USA) was used for clean reads mapping to generate the positioning information of reads on the reference genome. The mapped reads of each sample were assembled by StringTie. Reads mapped to each gene were counted by featureCounts. Differential gene expression was calculated by DESeq2 (Harvard School of Public Health, Boston, MA, USA). MT sample was used as a control for differentially expressed gene (DEG) analysis. The threshold of DEGs was judged by Padj ≤ 0.05 and |log2(FC)| > 1. The annotation analysis of DEGs was performed using clusterProfiler (V4.0 Southern Medical University, Guangzhou, China).

### 4.7. Gene Expression qRT-PCR Analysis

The total RNA of samples was extracted using the Quick RNA Isolation Kit. cDNA was synthesized using the Goldenstar^TM^ RT6 cDNA Synthesis Kit (Tsingke, Beijing, China). qRT-PCR primers were designed using Primer Premier 5.0 software (Table 2), with *CsTIP41* as a reference gene for light intensity treatment [70]. Quantitative amplification reactions were performed on the CFX96^TM^ Real-Time System (Bio-Rad, Hercules, CA, USA) using 2 × T5 Fast qPCR Mix (SYBR Green I) (Tsingke, Beijing, China) as the fluorescence reagent. The total volume of the quantitative amplification system was 20 μL: 10 μL 2 × T5 Fast qPCR Mix (SYBR Green I), 0.8 μL each of forward and reverse detection primers at a concentration of 10 μmol/L, 2 μL template cDNA, and 6.4 μL nuclease-free water. The quantitative amplification program was as follows: 1 min pre-denaturation at 95 °C; denaturation for 10 s at 95 °C; annealing and extension at 60 °C for 10 s; and 40 cycles. The gene relative expression levels were calculated using the Pfaffl method [71]. Samples were analyzed by qRT-PCR using three biological and three technical replicates.

### 4.8. Bioinformatics and Statistical Analysis

Heat maps were plotted using the Heatmapper website (http://www.heatmapper.ca/expression/ (accessed on 18 January 2023)). Histograms and line plots were plotted using Prism 8.0. Data were expressed as mean ± standard deviation (SD) and one-way analysis of variance (ANOVA) was performed based on SPSS 21 with the significance level *α* = 0.05.

## 5. Conclusions

The aim of this study was to investigate the effect of light intensity on the low-temperature adaptability of tea plants. Stronger light caused a decrease in chlorophyll and the yellowing of tea leaves. The photosynthetic activity of MT materials was optimal relative to the WT and ST materials according to the chlorophyll parameters *qP*, *Y(II)*, *ETR*, *NPQ*, and *Fv*/*Fm*. The activities of POD, SOD, CAT, APX, and PPO decreased with increasing light intensity, while MDA, soluble sugar, soluble protein, and relative conductivity increased. MT materials showed no significant damage after the frost resistance test, and the relative conductivity was the lowest. In contrast, both the ST and WT materials were subjected to frost damage. Strong light may have caused stress to the cells before frost through an increase in ROS, which led to browning on the leaf surface of ST materials after frost. Transcriptome sequencing showed that stronger light favors starch biosynthesis in carbon metabolism, while cellulose biosynthesis is enhanced in weaker light. Delayed tissue development and tenderness holding affect the frost intolerance of WT materials.

## Figures and Tables

**Figure 1 ijms-24-09852-f001:**
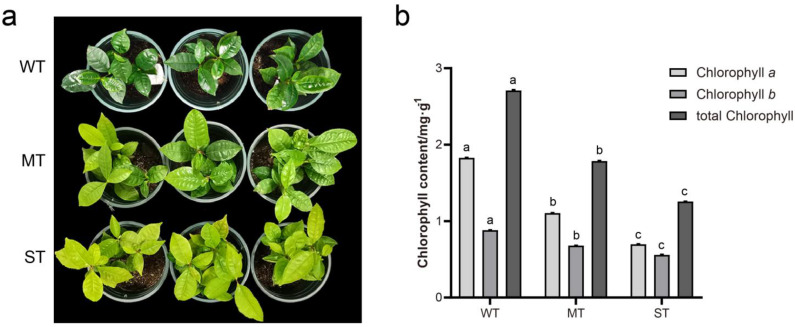
Observation of tea plant leaves under different light intensities. (**a**) Leaf phenotype; (**b**) chlorophyll content. Non−significant differences are marked with the same letter, while significant differences are marked with different letters. The significance level *α* = 0.05. WT, samples under weak light treatment (15 μmol·m^−2^·s^−1^). MT, samples under moderate light treatment (160 μmol·m^−2^·s^−1^). ST, samples under strong light treatment (240 μmol·m^−2^·s^−1^).

**Figure 2 ijms-24-09852-f002:**
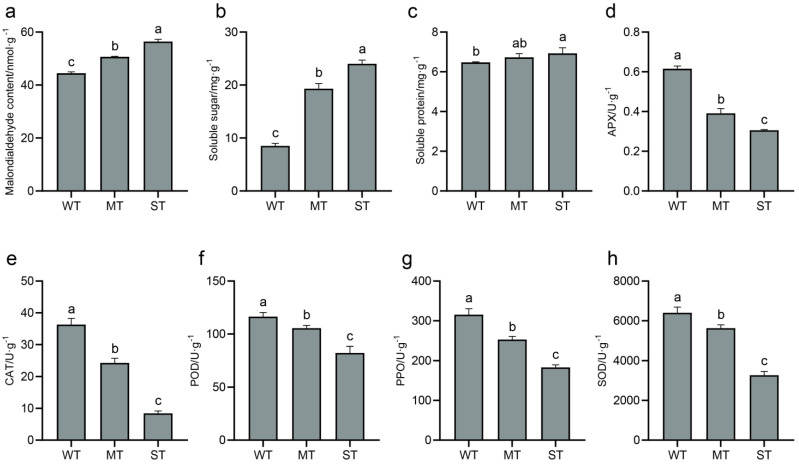
Low temperature tolerance indicators of tea plants. (**a**) Malondialdehyde (MDA) content; (**b**) soluble sugar content; (**c**) soluble protein content; (**d**) ascorbate peroxidase (APX) activity; (**e**) catalase (CAT) activity; (**f**) peroxidase (POD) activity; (**g**) polyphenol oxidase (PPO) activity; (**h**) superoxide dismutase (SOD) activity. Significance is indicated by superscript letters. Non−significant differences are marked with the same letter, while significant differences are marked with different letters. The significance level *α* = 0.05. WT, samples under weak light treatment (15 μmol·m^−2^·s^−1^). MT, samples under moderate light treatment (160 μmol·m^−2^·s^−1^). ST, samples under strong light treatment (240 μmol·m^−2^·s^−1^).

**Figure 3 ijms-24-09852-f003:**
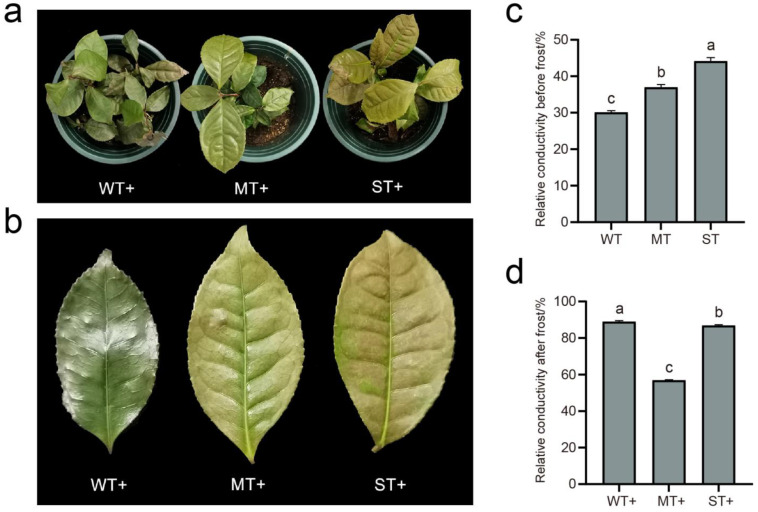
Frost treatment of tea plants. (**a**) Phenotypes of tea plants after frost treatment; (**b**) phenotypes of tea leaves after frost treatment; (**c**) relative conductivity of tea plants before frost treatment; (**d**) relative conductivity of tea plants after frost treatment. Significance is indicated by superscript letters. Non−significant differences are marked with the same letter, while significant differences are marked with different letters. The significance level *α* = 0.05. WT, samples under weak light treatment (15 μmol·m^−2^·s^−1^). MT, samples under moderate light treatment (160 μmol·m^−2^·s^−1^). ST, samples under strong light treatment (240 μmol·m^−2^·s^−1^). +, samples with frost treatment.

**Figure 4 ijms-24-09852-f004:**
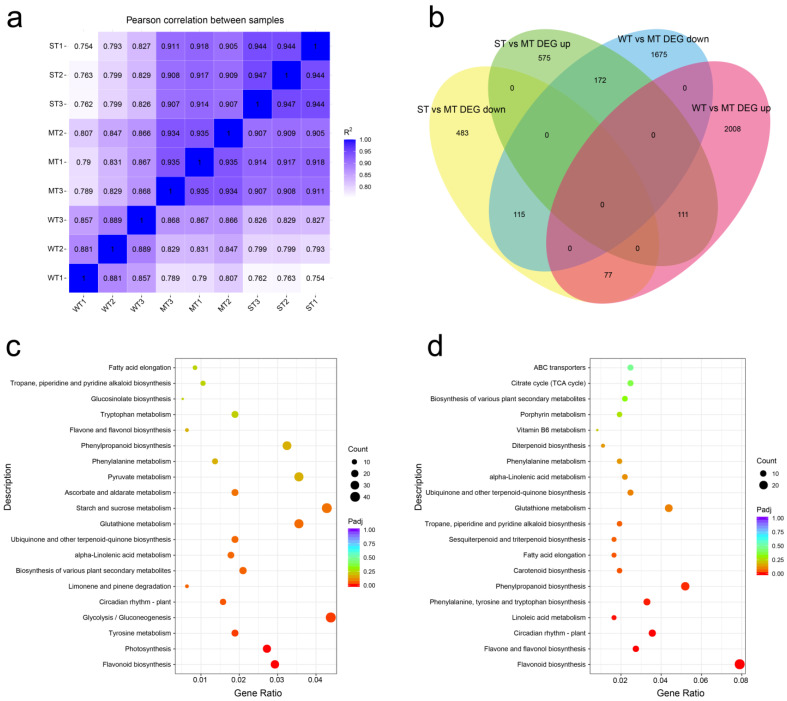
Transcriptome data analysis. (**a**) Comparison of transcriptional expression correlation between samples; (**b**) Venn diagram analysis of differentially expressed genes; (**c**) top 20 KEGG pathway matched with differentially expressed genes of WT vs. MT; (**d**) top 20 KEGG pathway matched with differentially expressed genes of ST vs. MT. WT, samples under weak light treatment (15 μmol·m^−2^·s^−1^). MT, samples under moderate light treatment (160 μmol·m^−2^·s^−1^). ST, samples under strong light treatment (240 μmol·m^−2^·s^−1^).

**Figure 5 ijms-24-09852-f005:**
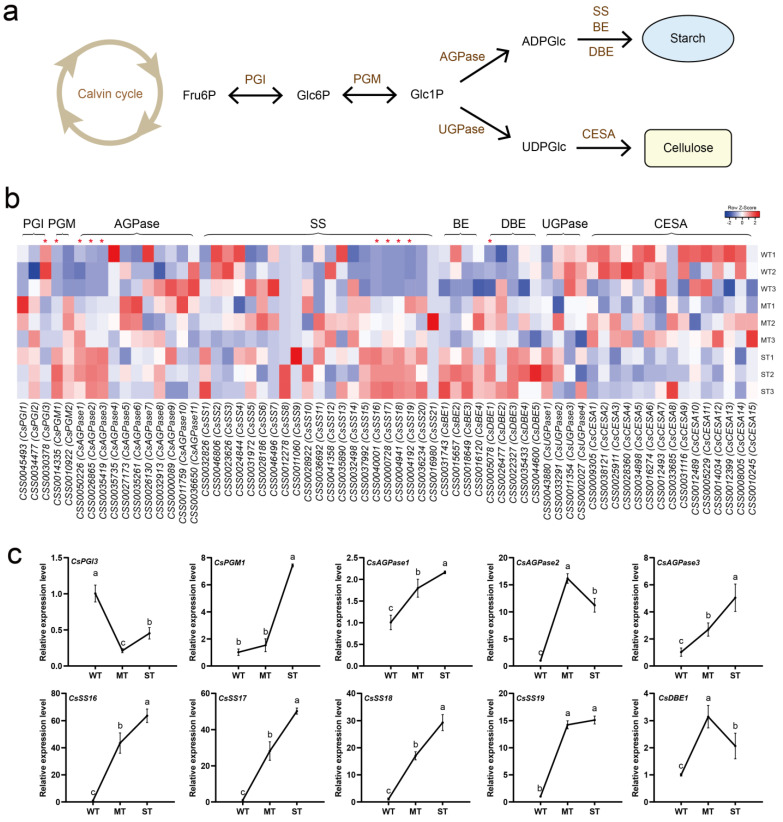
Light intensity-regulated starch and cellulose biosynthesis in tea plants. (**a**) Starch and cellulose biosynthetic pathways; (**b**) heatmap of gene expression in the starch and cellulose biosynthetic pathways; (**c**) DEG detection by qRT-PCR. PGI, phosphoglucose isomerase. PGM, phosphoglucomutase. AGPase, ADP-glucose pyrophosphorylase. SS, starch synthase. BE, branching enzyme. DBE, debranching enzyme. UGPase, UDP-glucose pyrophosphorylase. CESA, cellulose synthase A. Significance is indicated by superscript letters. Non−significant differences are marked with the same letter, while significant differences are marked with different letters. The significance level *α* = 0.05. *, differentially expressed genes. WT, samples under weak light treatment (15 μmol·m^−2^·s^−1^). MT, samples under moderate light treatment (160 μmol·m^−2^·s^−1^). ST, samples under strong light treatment (240 μmol·m^−2^·s^−1^).

**Figure 6 ijms-24-09852-f006:**
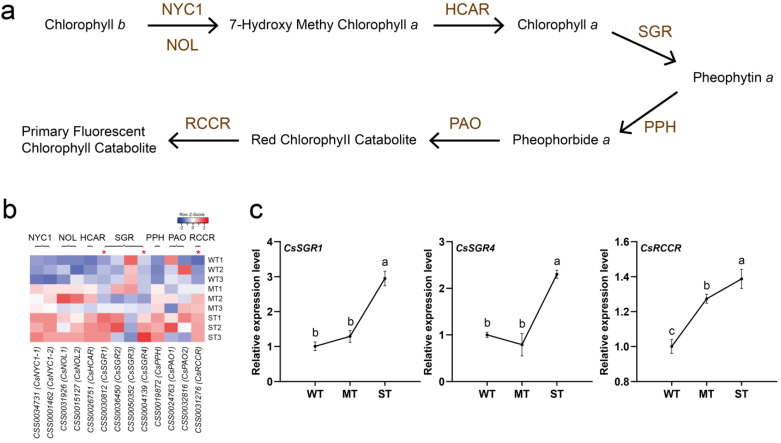
Light intensity-regulated chlorophyll degradation in tea plants. (**a**) The pathway of chlorophyll degradation. (**b**) Heatmap of gene expression in the pathway of chlorophyll degradation. (**c**) DEG detection by qRT-PCR. NYC1, NON-YELLOW COLORING 1. NOL, NYC1-Like. HCAR, 7-hydroxymethyl chlorophyll *a* reductase. SGR, STAY-GREEN. PPH, pheophytinase. PAO, pheophorbide *a* oxygenase. RCCR, red chlorophyll catabolite reductase. Significance is indicated by superscript letters. Non−significant differences are marked with the same letter, while significant differences are marked with different letters. The significance level *α* = 0.05. *, differentially expressed genes. WT, samples under weak light treatment (15 μmol·m^−2^·s^−1^). MT, samples under moderate light treatment (160 μmol·m^−2^·s^−1^). ST, samples under strong light treatment (240 μmol·m^−2^·s^−1^).

**Figure 7 ijms-24-09852-f007:**
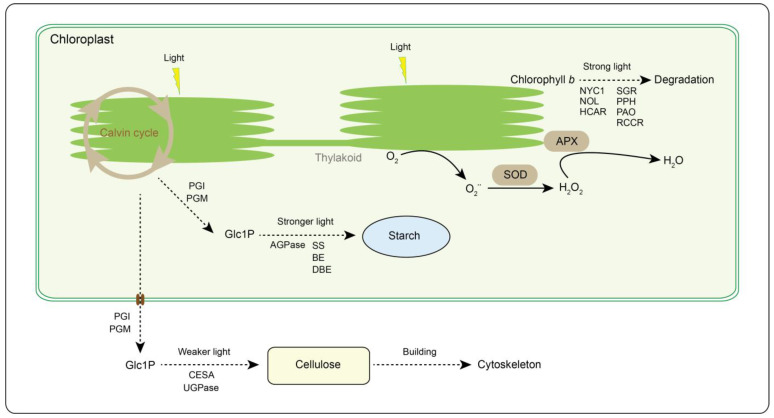
Light intensity response modes in tea plant. PGI, phosphoglucose isomerase. PGM, phosphoglucomutase. AGPase, ADP-glucose pyrophosphorylase. SS, starch synthase. BE, branching enzyme. DBE, debranching enzyme. UGPase, UDP-glucose pyrophosphorylase. CESA, cellulose synthase A. NYC1, NON-YELLOW COLORING 1. NOL, NYC1-Like. HCAR, 7-hydroxymethyl chlorophyll *a* reductase. SGR, STAY-GREEN. PPH, pheophytinase. PAO, pheophorbide *a* oxygenase. RCCR, red chlorophyll catabolite reductase. APX, ascorbate peroxidase. SOD, superoxide dismutase. Glc1P, glucose 1-phosphate. Solid and dotted lines represent one-step and multi-step responses, respectively.

**Table 1 ijms-24-09852-t001:** Chlorophyll fluorescence parameters of tea leaves under different light intensities.

Chlorophyll Fluorescence Parameters	WT	MT	ST
Maximum fluorescence, *Fm*	1.15 ± 0.05 ^a^	1.13 ± 0.06 ^a^	0.81 ± 0.05 ^b^
Variable fluorescence, *Fv*	0.89 ± 0.04 ^a^	0.85 ± 0.05 ^a^	0.54 ± 0.03 ^b^
Minimum fluorescence, *Fo*	0.26 ± 0.01 ^b^	0.28 ± 0.01 ^a^	0.26 ± 0.02 ^b^
Maximum photosynthetic quantum yield of PS II, *Fv/Fm*	0.78 ± 0.00 ^a^	0.75 ± 0.01 ^b^	0.67 ± 0.01 ^c^
Actual photosynthetic efficiency of PS II, *Y(II)*	0.24 ± 0.01 ^b^	0.27 ± 0.01 ^a^	0.25 ± 0.02 ^ab^
Photosynthetic electron transfer rate, *ETR*	14.05 ± 0.63 ^b^	16.02 ± 0.87 ^a^	15.07 ± 1.33 ^ab^
Non-photochemical quenching, *NPQ*	1.35 ± 0.18 ^ab^	1.51 ± 0.14 ^a^	1.23 ± 0.13 ^b^
Photochemical quenching, *qP*	0.40 ± 0.02 ^b^	0.50 ± 0.03 ^a^	0.53 ± 0.03 ^a^

The data in the table are mean ± standard deviation. Significance is indicated by superscript letters. Non−significant differences are marked with the same letter, while significant differences are marked with different letters. The significance level *α* = 0.05. WT, samples under weak light treatment (15 μmol·m^−2^·s^−1^). MT, samples under moderate light treatment (160 μmol·m^−2^·s^−1^). ST, samples under strong light treatment (240 μmol·m^−2^·s^−1^).

**Table 2 ijms-24-09852-t002:** Primer sequences used for qPT-PCR.

Gene Name	Upstream Primer Sequence (5’-3’)	Downstream Primer Sequence (5’-3’)
*CsPGI3*	GGAGCTATGGCGGAGATACCTTGAC	GCCTGAAACTTCGGCTCCATCTC
*CsPGM1*	TCTGGTGGTCTAAAGGGCGTTGC	GAGCCAAGCCAATACAGCCCAGA
*CsAGPase1*	GAGAAAGCTGCTGCAAACTACCCC	CGATGTCCTCCCAGTAGTCCTTGA
*CsAGPase2*	GAATAAGGGGGAGTCTGAAGAGCA	CTCAAATGATGGTGCCTCAAACG
*CsAGPase3*	CATCTCCCCGATTCTTGCCACC	GGGACCTTATCCTCAGCCAACA
*CsSS16*	AGAAGTTGGGTTGCCTGTGGACG	TGCCAGTGCCAAGAACTACAATC
*CsSS17*	TTGGATAACATCATTCGTAAGACCG	CGTCCACAGGCAACCCAACTTCT
*CsSS18*	GGCATAGATAAGGGTGTGGAATTG	CCAATCAAGGGGATGTTTGCGTC
*CsSS19*	GGTATTGTCAATGGCATGGATGTCC	CGTCCACAGGCAACCCAACTTCT
*CsDBE1*	TGCTGAACTTCGCAGGCTGTGGG	CACTGGCAAGGTCAAAGCGGAAA
*CsSGR1*	GGAAGAAAGTCCAGGAGAAAATGTC	GGAGTGAAAATATACCCAAACCAGA
*CsSGR4*	GAGTGGAAGGAAGTGAAAGGGGATA	CCAATGGCAACTCCTTTGAGAATAT
*CsRCCR*	CAGCAGGAGCACTGAACATAACGAG	GATGGAGAACAAGGTCTTTTCGAGG
*CsNYC1-1*	CGCCACTTCACTGCCACCGA	CTCAGACCTAAAGGACCTCATACGC
*CsTIP41*	TGGAGTTGGAAGTGGACGAGACCGA	CTCTGGAAAGTGGGATGTTTGAAGC

## Data Availability

Not applicable.

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
