# Peer review of "Light Intensity Regulates Low-Temperature Adaptability of Tea Plant through ROS Stress and Developmental Programs"

_ijms, 2023, doi:10.3390/ijms24129852_

Round 1
Reviewer 1 Report
Dear Editors,
The manuscript entitled "Light intensity regulates low temperature adaptability of tea plant through ROS stress and developmental programs” by Xin Zhang et al. is within the aim and scope of International Journal of Molecular Sciences. The authors have conducted experiments investigating the effect of light and temperature, alone or in combination, essential ecological factors of tea plant (Camellia sect. Thea) plant life. The research includes the essential roles of genes’ activation and regulation under the relevant responses. Abiotic stresses through several pathways, including the regulation of gene expression, are also discussed.
Overall, the manuscript deals with an interesting topic of high scientific significance. The manuscript is well-written. However, there are a few points of concern and I recommend minor revisions, according to the comments in the attached PDF file.

Reviewer 2 Report
In this study, the effect of light intensity on low temperature adaptability of tea plants 92 was assessed. We believe the ideas are clearly and accurately communicated without errors in spelling, grammar, and adequate word choice.
Main comment
In material and methods section authors should explain which statistical methods and software were used in results analysis.
General comment
Figures and tables: Please explain the meaning of WT, MT and ST in figures and tables captions.
